# Overexpressing *GmCGS2* Improves Total Amino Acid and Protein Content in Soybean Seed

**DOI:** 10.3390/ijms241814125

**Published:** 2023-09-15

**Authors:** Yuchen Zhang, Qingyu Wang, Yajing Liu, Shuo Dong, Yongqiang Zhang, Youcheng Zhu, Yu Tian, Jingwen Li, Zhuoyi Wang, Ying Wang, Fan Yan

**Affiliations:** College of Plant Science, Jilin University, Changchun 130012, China

**Keywords:** cystathionine γ-synthase, *Glycine max*, methionine biosynthesis, soybean protein, sulfur-containing amino acid, methionine

## Abstract

Soybean (*Glycine max* (L.) Merr.) is an important source of plant protein, the nutritional quality of which is considerably affected by the content of the sulfur-containing amino acid, methionine (Met). To improve the quality of soybean protein and increase the Met content in seeds, soybean cystathionine γ-synthase 2 (*GmCGS2*), the first unique enzyme in Met biosynthesis, was overexpressed in the soybean cultivar “Jack”, producing three transgenic lines (OE3, OE4, and OE10). We detected a considerable increase in the content of free Met and other free amino acids in the developing seeds of the three transgenic lines at the 15th and 75th days after flowering (15D and 75D). In addition, transcriptome analysis showed that the expression of genes related to Met biosynthesis from the aspartate-family pathway and S-methyl Met cycle was promoted in developing green seeds of OE10. Ultimately, the accumulation of total amino acids and soluble proteins in transgenic mature seeds was promoted. Altogether, these results indicated that *GmCGS2* plays an important role in Met biosynthesis, by providing a basis for improving the nutritional quality of soybean seeds.

## 1. Introduction

Approximately 70% of all plant-based protein is derived from soybeans, which are cultivated worldwide [1]. Soybeans are considered a source of complete plant protein because they contain all nine essential amino acids. However, the content of sulfur-containing amino acids, that is, methionine (Met) and cysteine (Cys), in soybean seed protein is relatively low [2]. Sulfur-containing amino acids are essential amino acids that cannot be synthesized by nonruminants; thus, they must be obtained through the diet. These amino acids are important for human health and animal growth and development [3]. As the main sulfur-containing amino acid, Met can be transformed into Cys by animals, satisfying their need for these two amino acids [4]. As a basic metabolite in plant cells, Met also plays a key role in a variety of cell functions. For example, it is a component of protein synthesis that initiates mRNA translation [5]. Moreover, Met may participate in the regulation of ethylene and polyamine biosynthesis through SAM (a major methyl donor involved in a series of biological processes in plants during seed germination), which significantly affects seed germination and seedling growth [5,6]. Met is also involved in regulating the accumulation of seed storage proteins. For instance, exogenous Met added into the culture medium of immature cotyledons led to an obvious increase in the abundance of Met-rich 11S-proteins, whereas it decreased the abundance of Met-poor 7S β-subunit proteins, which are the main proteins of soybean [7]. Other studies have demonstrated that overexpressing AtD-CGS soybean seeds with higher Met content showed both higher total protein and amino acid content; however, the precise mechanism still needs further research [4].

Scientists have attempted to increase the Met content of seed protein to support its role in nutrient supply and seed germination and development. Utilizing traditional plant breeding methods and mutant selection was unsuccessful in achieving this goal as these efforts resulted in abnormal crop morphology and reduced yields [8,9]. From the perspective of seed storage proteins, increasing the content of native sink proteins rich in sulfur amino acids and heterologous seed proteins, or reducing storage proteins lacking sulfur-containing amino acids have also been attempted to improve the content of Met in seeds [10,11,12,13,14]. Despite the expression of a new Met-rich recombinant seed storage protein in kidney beans, no increase was detected in the total Met level in the seeds [15]. In addition, owing to the limitation of Met availability during the development of soybean seeds, increasing the Met content in seeds by introducing a Met-rich protein “library” has often been accompanied with a reduction in other endogenous sulfur-rich proteins and sulfur compounds [7,14,16,17]. Recent efforts to promote the synthesis of soluble Met in plants have focused on the direct manipulation of the Met biosynthetic pathway, resulting in improving the content of Met in seeds. Met can be synthesized de novo in seeds via the aspartate-family pathway, or in nonseed tissues via the *S*-methyl Met (SMM) cycle, where Met is produced in vegetative tissues and transported to developing seeds through transformation into SMM [18,19]. Owing to the uncertain contribution of the SMM cycle to Met metabolism in seeds, Met might be synthesized mainly through the aspartate-family pathway in legume seeds [20,21]. Cystathionine γ-synthase (CGS) is the first unique enzyme in Met biosynthesis that combines the carbon–amino skeleton from aspartic acid with the sulfur moiety from Cys, which is the first step in the Met biosynthesis pathway [22,23].

Constitutive overexpression of full-length *Arabidopsis thaliana* CGS (*AtCGS*) in tobacco, Arabidopsis, and alfalfa has successfully increased the Met content in transgenic leaves, suggesting that CGS is a key rate-limiting enzyme in the biosynthesis of Met [21,24,25]. However, overexpression of CGS in seeds has led to different results. For example, after overexpressing a feedback-insensitive form of *AtCGS* (*AtD-CGS*) in soybean, Arabidopsis, and tobacco, the levels of soluble and protein-bound Met in the seeds were substantially increased [4,26,27,28]. However, no increase in the Met content was observed in transgenic adzuki beans overexpressing *AtCGS1* (cloned from the Arabidopsis *mto1-1* mutant) or in transgenic potatoes overexpressing *Sulfolobus tokodaii* CGS (*StCGS*) [29,30]. Moreover, the total Met content was also considerably increased in transgenic soybean seeds constitutively overexpressing *AtD-CGS* [31]. These results suggested that overexpressing CGS may have different effects on Met biosynthesis among different species and that the CGS gene family may have evolved over time. To clarify the mechanisms of regulation of soybean CGS on Met biosynthesis, we overexpressed soybean CGS 2 (*GmCGS2*) in the soybean variety “Jack” and carried out transcriptome sequencing analysis of seeds at different developmental stages. Furthermore, we examined the effect of gene manipulation on the content of amino acids and proteins in developing and mature soybean seeds. The results of this study provide a theoretical basis for improving the nutritional quality of soybean.

## 2. Results

### 2.1. Expression of GmCGS2 Was Significantly Higher Than that of GmCGS1 at All Seed Developmental Stages

We obtained the protein amino acid sequences encoded by two homologous soybean genes, *GmCGS1* (Glyma. 18G261600) and *GmCGS2* (Glyma. 09G235400), using the NCBI Protein Blast Local Alignment Search Tool according to the amino acid sequence of *AtCGS1* (At3G01120). Phylogenetic analysis of the amino acid sequences of CGS proteins from *A. thaliana* (Arabidopsis), *Zea mays* (maize), *Oryza sativa* (rice), *Solanum tuberosum* (potato), *Nicotiana tabacum* (tobacco), and soybean showed that the two soybean and Arabidopsis CGS proteins were closely related (Appendix A). Both soybean CGS proteins contain a Cys/Met metabolism PyrdxlP-dependent enzyme domain similar to that of AtCGS1, while the amino acid sequence of AtCGS1 was 71.33% and 71.38% similar to that of GmCGS1 and GmCGS2, respectively (Appendix A and Figure 1A). The amino acid sequence similarity between GmCGS1 and GmCGS2 was 97.76%, indicating high conservation (Figure 1A). To investigate the subcellular localization of GmCGS1 and GmCGS2, we constructed GmCGS1-GFP and GmCGS2-GFP fusion plasmids and transiently expressed them in tobacco leaves. Confocal microscopy revealed a fluorescence signal in the lower epidermal cells, indicating the colocalization of GmCGS1-GFP and GmCGS2-GFP with chloroplast autofluorescence. Whereas, fluorescence from the empty vector-GFP was localized to the cytoplasm and nucleus (Figure 1D). Therefore, we concluded that both GmCGS1 and GmCGS2 were located in chloroplasts.

Previous studies have indicated that seed development in angiosperms occurs in three overlapping stages: (1) tissue differentiation, (2) grain filling, and (3) seed drying. The synthesis and accumulation of nutrients, such as amino acids and storage proteins, mainly occur during the grain-filling stage [32,33]. Thus, to understand the expression patterns of GmCGS1 and GmCGS2 during different developmental stages in soybean, we used developing seeds at 15D, 25D, 35D, 45D, 55D, 65D, and 75D for qRT-PCR analysis (Figure 1C). These samples contained the developmental stages from R5 (beginning seed) to R7 (beginning maturity) [34].

The expression levels of these two genes were influenced significantly by the developmental stages (F(6, 72) = 22.83, *p* < 0.0001), gene types (F(1, 72) = 775.1, *p* < 0.0001), and their interactive effect (F(6, 72) = 6.805, *p* < 0.0001). Moreover, the correlation between the expression levels of GmCGS1 and GmCGS2 in developing seeds at 15–75D was analyzed with Pearson’s correlation analysis, which showed that the expression of GmCGS1 and GmCGS2 exhibited similar change trends (Pearson’s correlation coefficients = 0.89, *p* = 0.00674). The expression levels of GmCGS1 and GmCGS2 were first decreased by 79.6% (*p* < 0.0001) and 35.5% (*p* < 0.0001), respectively, from 15D to 55D, then increased by 74.6% (*p* = 0.9020) and 33.1% (*p* < 0.0001), respectively, at 65D, and finally decreased again by 22% (*p* = 0.9972) and 7.4% (*p* < 0.0001), respectively, at 75D (Figure 1B). This result suggested that despite the highly similar subcellular localization, amino acid sequences, and expression patterns shared by GmCGS1 and GmCGS2, we noticed that the level of expression of *GmCGS2* was significantly higher than that of *GmCGS1* at all developmental stages. Therefore, we chose *GmCGS2* for further experimental analysis.

### 2.2. Transgenic Soybean Exhibited Elevated Levels of CGS during Seed Development

To further explore the regulatory role of *GmCGS2* in Met biosynthesis in soybean seeds, we overexpressed *GmCGS2* in the soybean variety “Jack” and identified three independent transformation events (OE3, OE4, and OE10) using PCR and PAT test strips. To clarify the introduction efficiency of *GmCGS2* in these transgenic events, we stripped the developing pods from WT and transgenic lines at 15D, 25D, 35D, 45D, 55D, 65D, and 75D and used the seeds as samples for qRT-PCR analysis. Our qRT-PCR analysis revealed a significant increase in the levels of expression of *GmCGS2* in developing seeds at 15D, 45D, and 65D in OE3, in developing seeds at 15D, 25D, 35D, and 45D in OE4, and in developing seeds at all stages in OE10. In general, the levels of expression of *GmCGS2* showed a consistent increasing trend in the seeds of the three transgenic lines at early developing stages and a smaller range of increase in OE3 and OE10 seeds at late stages (Appendix A).

### 2.3. Overexpression of GmCGS2 Increased the Content of Free Met and Total Free Amino Acids

To explore whether overexpression of *GmCGS2* leads to an increase in the concentration of Met in seeds, the content of free amino acids in developing seeds from the three transgenic lines and the WT at 15D (initial stage of grain development), 55D (stage of complete grain bulging), and 75D (later stage of grain maturity) was quantified by UHPLC-MS/MS. The results showed that the overexpression of *GmCGS2* increased the content of free Met significantly by 123*%* (*p* < 0.0001), 50*%* (*p* = 0.0034), and 41*%* (*p* = 0.0052) in OE3, OE4, and OE10 developing seeds, respectively, at 15D (F = 66.28, *p* < 0.0001), and by 44*%* (*p* = 0.0346) and 34*%* (*p* = 0.0377) in OE3 and OE10 developing seeds, respectively, at 75D (F = 6.499, *p* = 0.0354), compared with that of WT developing seeds (Figure 2). In contrast, we detected that the levels of Met in developing seeds at 55D (F = 7.769, *p* = 0.0382) were not significantly different between the three transgenic lines and the WT (Figure 2A–C). These results indicated that *GmCGS2* is involved in the regulation of Met biosynthesis in soybeans.

Moreover, we observed that overexpressing *GmCGS2* substantially increased the levels of most other free amino acids in the three transgenic lines at 15D compared with those in WT developing seeds, resulting in 42.8*%* (*p* < 0.0001), 11*%* (*p* = 0.0002), and 27.5*%* (*p* < 0.0001) significant accumulation of total free amino acids in OE3, OE4, and OE10, respectively (F = 750.9, *p* < 0.0001) (Figure 2D). A similar pattern was observed at 75D, with significant increases in the content of total free amino acids of 73.5*%* (*p* = 0.0003), 35.3*%* (*p* = 0.0133), and 27.1*%* (*p* = 0.0259) in OE3, OE4, and OE10, respectively (F = 26.69, *p* = 0.0007) (Figure 2F). However, the change in the content of total free amino acids differed among the three transgenic lines at 55D: no significant change in OE3, but a significant increase of 27.4*%* (*p* = 0.0044) in OE4, and a significant decrease of 44*%* (*p* = 0.0007) in OE10 (F = 131.8, *p* = 0.0002) (Figure 2E).

### 2.4. Overexpression of GmCGS2 Increased the Content of Total Protein-Bound Amino Acids and Soluble Protein but Not Protein-Bound Met

To investigate whether the higher content of free Met in developing seeds leads to an increased amount of protein-bound Met in mature seeds, we evaluated the content of protein-bound amino acids and total protein during grain development. The results revealed that the content of protein-bound Met in the transgenic line seeds was not affected compared with that in the WT seeds (Figure 3C). However, the content of the majority of other protein-bound amino acids in seeds was significantly increased, resulting in an increase in the total protein-bound amino acids of 5.7*%* (*p* = 0.0098), 7.7*%* (*p* = 0.0017), and 15.3*%* (*p* < 0.0001) in transgenic lines OE3, OE4, and OE10, respectively, compared with that in the WT (F = 39.17, *p* < 0.0001) (Figure 3A,C).

To determine whether a higher content of protein-bound amino acids promotes the accumulation of soluble proteins in seeds, total soluble protein levels in WT and transgenic line seeds were analyzed. We found that the content of total soluble protein was significantly increased by 4.7*%* (*p* = 0.0049), 4.6*%* (*p* = 0.0057), and 9.5*%* (*p* < 0.0001) in transgenic lines OE3, OE4, and OE10, respectively, compared with that in the WT (F = 15.4, *p* < 0.0001) (Figure 3B).

### 2.5. Global Transcriptome Responses of Developing Soybean Seeds after Overexpression of GmCGS2

Eighteen libraries were constructed and sequenced to further explore the mechanism of *GmCGS2* in regulating Met biosynthesis, and 841 million clean reads were obtained, with 40–52 million clean reads generated per library. The distribution of the GC content and sequencing error rate of raw reads are shown in Appendix A. Approximately 95*%* of the clean reads were mapped to 58,146 genes in the reference *Glycine max* genome. The total, unique, and multimapped reads are summarized in Appendix A, while all RNA-seq data are summarized in Appendix A.

A total of 7281 differentially expressed genes were identified using DESeq2 (*p* ≤ 0.05 and |log2(fold change) | ≥ 1), between WT and OE10 developing seeds at 15D, 55D, and 75D, including 3673 upregulated and 3608 downregulated genes (Figure 4A). The results showed that the number of DEGs between the WT and OE10 developing seeds at the three stages were 4808, 1376, and 1097, respectively, decreasing in number with the development of the soybean seeds (Figure 4A). PCA (principal component analysis) showed a clear separation between different samples, especially at different immature seed developmental stages. The PCA results also displayed good repeatability for the same treatment sample (Figure 4B). The reliability of RNA-seq data was validated by qRT-PCR (Appendix A).

All DEGs between the WT and OE10 developing seeds at the three developmental stages were linearly clustered using the H-cluster method, and five clustering results with significant change trends (upregulated or downregulated) were selected for subsequent analysis (Figure 4C). The results showed that 6020 DEGs were enriched in cluster 1, in which DEGs showed an upward trend in expression in OE10 developing seeds at 15D compared with those of the WT, and a downward trend in expression in WT and OE10 developing seeds at 55D compared with the same type of developing seed at 15D. In addition, we annotated the DEGs in each cluster using the KEGG database. A total of 119 metabolic pathways were jointly annotated in cluster 1, of which 21 metabolic pathways were significantly enriched (*p* < 0.05). The enriched metabolic pathways included amino acid metabolic pathways (ko00270, ko00280, ko00380, ko00410, ko00460, ko00480, and ko00940), lipid and glucose metabolic pathways (ko00592, ko00564, ko00561, ko00062, ko00500, and ko00052), synthesis of plant hormones and secondary metabolites (ko00903, ko00941, ko00905, and ko00908), and other metabolic pathways (ko02010, ko04016, ko04626, and ko00196). A total of 336 DEGs were enriched in cluster 2 and annotated with 46 metabolic pathways, of which five metabolic pathways were significantly enriched, including the synthesis of some secondary metabolites (ko00941 and ko00909) and amino acid and glucose metabolism (ko00480, ko00500, ko00052, and ko00010). In total, 16, 17, and 28 DEGs were enriched in clusters 3, 4, and 5, respectively, which were annotated with two, three, and five metabolic pathways, respectively. Six metabolic pathways were significantly enriched, including some cofactors and lipid metabolism (ko00860, ko00591, and ko00073), synthesis of amino acids and secondary metabolites (ko00941 and ko00220), and protein processing (ko04141) (Figure 4C).

### 2.6. Overexpression of GmCGS2 Positively Regulated the Aspartate-Family Pathway and SMM Cycling

By searching the KEGG pathway database with the transcriptome results, we found that 47 DEGs were enriched in the Cys and Met metabolism pathway. We further investigated the molecular mechanism of the Met biosynthesis response to the overexpression of *GmCGS2* during the development of soybean seeds based on the content of free amino acids and transcriptome results. According to RNA-seq analysis, the level of expression of *GmCGS1* was not affected in developing seeds of OE10 at 15D, 55D, and 75D compared with that of the WT. Accordingly, we assumed that the changes in the levels of expression of DEGs and content of free amino acids involved in the pathway likely stemmed from the overexpression of *GmCGS2* (Figure 5A).

In OE10 developing seeds at 15D, the content of free aspartate (Asp) and the levels of expression of genes novel.158 and Glyma.18g002900 encoding bifunctional aspartokinase–homoserine dehydrogenase (AK–HSDH), which catalyzes the degradation of Asp, were significantly increased compared with those in the WT (Figure 5A) [35]. Notably, we found that the expression of the three genes encoding AK–HSDH, novel.158, Glyma.18g002900, and Glyma.13g048300, showed a 3.12-fold increase, a 4.32-fold increase, and no apparent change, respectively, while the transcript levels of Glyma.18g002900 and Glyma.13g048300 were quite low (Figure 5B). Therefore, it is likely that novel.158 plays a crucial role in the degradation of Asp. However, the expression of the gene encoding threonine synthase (TS), the key enzyme in the threonine synthesis pathway, was not affected (Figure 5A). Whereas, the expression of genes encoding enzymes related to Cys biosynthesis, including serine acetyltransferase (SAT) and cysteine synthase (OAS-TL), was significantly induced. We also noticed that the expression of the gene encoding homocysteine-S-methyltransferase (HMT), which is associated with the SMM pathway, was also significantly upregulated in the transgenic developing seeds at 15D. Therefore, we concluded that the overexpression of *GmCGS2* promoted the biosynthesis of Met via the aspartate-family pathway and SMM cycle, thereby increasing the content of free Met in developing seeds at 15D.

In OE10 developing seeds, somewhat smaller changes in the aspartate-family pathway were observed at 75D than those observed at 15D compared with the WT; however, the expression of HMT was not substantially promoted in OE10 at 75D (Figure 5A). Thus, the increased accumulation of free Met during this period might be attributed to the activated aspartate-family pathway. We found that the content of free Asp and serine (Ser) did not differ markedly in OE10 at 55D compared with that in the WT. Although the levels of expression of OAS-TL and novel.158 were considerably upregulated, no marked increase was observed in the level of free Met in OE10 at 55D (Figure 5A). In addition, there was no substantial change in the level of expression of Met-γ-lyase (MGL) in OE10 at 15D and 75D compared with that in WT, indicating that the increase in the content of free Met in OE10 did not promote the degradation of Met (Figure 5B). Notably, we noticed that in OE10 at 55D, which was characterized by no marked change in the content of Met, the level of expression of MGL was considerably decreased compared with that of the WT, hence inhibiting the degradation of Met (Figure 5A). Although the levels of expression of S-adenosylmethionine synthetase (SAMs) were not substantially changed in transgenic seeds at the three stages compared with those in the WT, the transcription of enzyme genes for ethylene and S-methyl-5′-thioadenosine synthesis, ACC synthase (ACS), ACC oxidase (ACO), S-adenosylmethionine decarboxylase proenzyme (SPE), and spermidine synthase (SPDS) was considerably promoted in transgenic seeds at 15D (Figure 5A). It is likely that the accumulation of free Met provided more substrate for ethylene and S-methyl-5′-thioadenosine biosynthesis, thereby promoting the synthesis of ethylene and S-methyl-5′-thioadenosine in developing seeds at 15D.

### 2.7. GmCGS2 Expression May Be Related to Transcription Factors Responding to Biotic and Abiotic Stressors

The key elements regulating gene expression are transcription factors. A total of 411 DEGs annotated as transcription factors were identified in the three comparison combinations. These transcription factors were grouped into subclasses, such as MYB, AP2, and bHLH (Figure 6A). To further understand the upstream regulatory mechanism related to the expression of *GmCGS2*, we performed a Pearson’s correlation analysis for the levels of gene expression of differentially expressed transcription factors and *GmCGS2* in WT and OE10 developing seeds at 15D, 55D, and 75D (Figure 6B). To this end, we selected the transcription factors that were significantly correlated with *GmCGS2* (Pearson’s correlation coefficients ≥ 0.85) (Figure 6C). Notably, we observed that most of the selected transcription factors were related to plant responses to biotic and abiotic stress (Figure 6D).

## 3. Discussion

Sufficient Met supplementation is necessary for improving the quality and nutritional value of protein from soybeans. Regulating the function of the first key enzyme in the Met biosynthesis pathway is the most promising target for increasing the content of Met, with the fewest side effects [51]. In this study, we demonstrated that *GmCGS2*, a homologous gene of *AtCGS1*, encodes CGS2 that participates in the biosynthesis of Met in soybeans. In addition, transgenic seeds overexpressing *GmCGS2* had higher contents of total amino acids and protein.

In this study, the higher levels of expression of GmCGS2 were mainly reflected in developing seeds at the early stages in the WT. The increase in the level of expression of GmCGS2 and contents of free Met and other free amino acids was also mainly reflected at the same stages of the transgenic seeds, leading to an increase in the contents of total amino acids and soluble protein in transgenic mature seeds (Appendix A, Figure 2 and Figure 3). Studies have shown that most sulfur metabolic enzymes are highly abundant at the early stages of seed development compared with the mid- and late stages, and this was accompanied by an increased synthesis of sulfur-containing amino acids [52,53,54]. In addition, many studies suggested a positive correlation between the level of expression of *AtCGS* and accumulation of free Met and other free amino acids [3,4,27]. Besides, the synthesis of storage protein begins at the early stages of developing seeds, which is consistent with the rapidly decreasing content of free amino acids after this stage [55,56]. Therefore, we considered that GmCGS2 mainly functions in developing seeds at early stages, and the moderate increase in the transcript level of GmCGS2 and content of free Met in the late developmental stages was likely due to the additional needs of storage protein synthesis (Appendix A and Figure 2) [54].

Our study showed that an increase in the content of free Met did not result in an increase in protein-bound Met in transgenic mature seeds (Figure 3C). An inadequate increase in the content of free Met in developing seeds at the early developmental stages is the likely cause of the unchanged accumulation of protein-bound Met in transgenic seeds. In a previous study, transgenic soybean lines with higher levels of *AtD-CGS* overexpression and content of free Met in green developing seeds exhibited a significant increase in the content of protein-bound Met in mature seeds, whereas those with lower levels of *AtD-CGS* and content of free Met were unchanged [4]. In addition, the inadequate increase in the levels of free Met in transgenic developing seeds may also be related to our selection of the full-length *GmCGS2* gene for the experiments. Many studies have shown that in Arabidopsis, the expression of *AtCGS* is involved in the feedback regulation of mRNA stability, which is autoregulated by the cytosolic concentration of SAM, a direct metabolite of Met [57,58]. A region of approximately 90 amino acids in the N-terminus of *AtCGS*, which is not necessary for the catalytic activity of the enzyme, is mainly responsible for this feedback regulation mechanism [59]. Transgenic plants engineered to constitutively express *AtD-CGS* accumulated more Met than those expressing full-length feedback-sensitive *AtCGS* [21,59]. In addition, both seed-specific and constitutive overexpression of insensitive *AtD-CGS* resulted in higher levels of both soluble and protein-incorporated Met in soybean transgenic lines compared with those in the control [4,31]. Similar observations were also made in Arabidopsis and tobacco plants that specifically overexpressed *AtD-CGS* and *EcAK* in seeds, respectively [26,28]. Our study showed that the increased levels of expression of SPE and ACS in the transgenic lines contributed to a higher concentration of SAM (Figure 4A). This higher concentration likely initiates a feedback inhibition regulation on the transcript level of CGS, resulting in reduced accumulation of free Met in our transgenic lines, ultimately limiting the increase in the content of protein-bound Met in the mature seeds of the transgenic lines.

To further analyze the molecular mechanism by which *GmCGS2* regulates Met biosynthesis during grain development, RNA-seq analysis was performed. According to previous studies, Met in soybean seeds can be synthesized de novo through the aspartate-family pathway or the SMM pathway in nonseed tissues [18,19]. Although the contribution of the aspartate-family pathway to Met biosynthesis in plant seeds has been widely reported, the contribution of the SMM cycle to the Met biosynthesis in plant seeds remains unclear. In feeding experiments using transgenic Arabidopsis plants expressing RNAi::AtCGS and transgenic soybean plants overexpressing AtD-CGS constitutively, the SMM cycle was recognized as contributing to the increase in the content of Met in seeds [31,60]. However, some studies contradicted the role of the SMM cycle in plant seeds, as they showed that the content of Met in Arabidopsis and maize mmt mutants with inactivation of the gene encoding Met S-methyltransferase was unaltered compared with that in the WT [61]. Moreover, although the content of Met and SMM was substantially increased in the leaves of transgenic tobacco overexpressing *AtD-CGS* together with AK in the feedback-insensitive form, the content of Met in the seeds increased only slightly [21]. Therefore, we inferred that the different effects of the overexpression of *GmCGS2* on two Met biosynthesis pathways in transgenic seeds at different developmental stages were likely due to the main contribution of the aspartic acid pathway to the accumulation of Met in soybeans, with the SMM cycle potentially representing a temporary complementary strategy for the increased supply of Met in the early stages of seed development in our transgenic lines.

Based on the results of RNA-seq analysis, we found that the transcription factors that were significantly correlated (Pearson’s correlation coefficients ≥ 0.85) with *GmCGS2* mainly related to plant responses to biotic and abiotic stressors; for example, CabHLH79 and CsbHLH18 enhance cold resistance by regulating the homeostasis of reactive oxygen species and by directly regulating the antioxidant gene in pepper and sweet orange, respectively [38,45] (Figure 6D). In addition, SAM, the main catabolite of Met, is involved in regulating ethylene synthesis, which is an important plant hormone response to biotic and abiotic stress [62,63]. Furthermore, Met has been demonstrated to play a crucial role in oxidative stress. Transgenic tobacco plants with higher levels of Met also showed higher levels of stress-related metabolites after overexpressing *AtD-CGS* [64]. In Arabidopsis overexpressing *AtD-CGS*, seeds that accumulated more Met exhibited higher germination rates when grown under salt and osmotic stress [26]. Overall, we believe that transcription factors related to stress may be involved in regulating the expression of *GmCGS2*. However, the relationship between the Met pathway and stress response remains unclear; therefore, additional research is required to determine the level of expression of *GmCGS* and the corresponding transcription factors under various stressors.

In summary, we identified the important role that *GmCGS2* plays in the biosynthesis of Met in soybean seeds. In addition, we found that overexpression of *GmCGS2* promoted the transcription of genes encoding upstream enzymes and the synthesis of substrates in the aspartate-family pathway, resulting in a substantial increase in the content of total amino acids and soluble protein in transgenic mature seeds. Through RNA-seq, we also determined that the expression of *GmCGS2* was highly correlated with transcription factors related to stress resistance, which has not been previously reported. The results of this study provide a theoretical basis and gene resources for improving the nutritional quality of soybean germplasm.

## 4. Materials and Methods

### 4.1. Phylogenetic Analysis

The DNA sequences and encoded protein sequences of two *GmCGS* genes were obtained using the NCBI Protein Blast tool (http://blast.ncbi.nlm.nih.gov/Blast.cgi, accessed on 19 March 2021). Protein sequence similarity among GmCGS1, GmCGS2, and AtCGS1 was analyzed using DNAMAN, and the results were presented using GENEDOC. Conserved functional domains were identified using SMART (http://smart.embl-heidelberg.de/, accessed on 19 March 2021), and the diagram of the protein sequence was drawn using Illustrator for Biological Sequences (http://ibs.biocuckoo.org/download.php, accessed on 19 March 2021). The amino acid sequences were aligned using ClustalX2, and the neighbor-joining phylogenetic tree was constructed using MEGA6 with the Neighbor-Joining method and a p-distance metric bootstrap value of 1000 [65]. The Interactive Tree of Life (iTOl) online tool (https://itol.embl.de/, accessed on 3 April 2021) was used for visualization of the phylogenetic tree.

### 4.2. Plants and Growth Conditions

Two sources of *Glycine max* (L.) germplasm, Jilin 26 (a high-protein soybean variety) and Jack, were used as the cloning target and genetic background for transformation, respectively. Seedlings were grown in a climate-controlled chamber (24 ± 2 °C, 16 h:8 h light:dark, 60% humidity), along with T0 and T1 plants. T2 and T3 plants were planted at the Safe Release Base of Genetically Modified Crops, Jilin University, China (N 43°55′22.6344″, E 125°16′32.9952″). As T3 plants grew, 15-, 25-, 35-, 45-, 55-, 65-, and 75-day developing seeds after flowering of wild-type (WT) and three transgenic lines were harvested, immediately frozen in liquid nitrogen, and stored at −80 °C for further analysis.

*Nicotiana benthamiana* was used for transient transfection, and four-week-old *N. benthamiana* leaves were used for further experimentation.

### 4.3. Quantitative Reverse-Transcription (qRT)-PCR Analysis

For qRT-PCR, total RNA from 15-, 25-, 35-, 45-, 55-, 65-, and 75-day developing seeds after flowering of WT and three transgenic lines was extracted with Transzol (Transgen, Beijing, China), according to the manufacturer’s instructions. cDNA Synthesis Super Mix with One-Step gDNA Removal (Transgen, China) was used for cDNA synthesis. qRT-PCR assays were performed on a Bio-Rad CFX Manager System (ThermoFisher Scientific, Shanghai, China) using SYBR Green I mix (Transgen, China). The results were analyzed using Bio-Rad CFX Manager 3.0 software, and the relative expression level was calculated using the 2^−ΔΔCt^ method [66]. Two technical replicates of three biological replicates were analyzed. An internal reference gene, soybean Actin, was chosen to standardize the data. The specific primers used for qRT-PCR analysis are listed in Appendix A.

### 4.4. Subcellular Localization Analysis

To confirm the subcellular localization of GmCGS2 in a transient tobacco expression system, the coding sequences of GmCGS2 and green fluorescent protein (GFP) were fused using overlap extension PCR, and the fusion fragment was introduced into the BamH1 restriction sites of the binary vector pCAMBIA3301 with the cauliflower mosaic virus (CaMV) 35S promoter; the primers are listed in Appendix A. Agrobacterium strain EHA105 containing 35S: GmCGS2-GFP and 35S: GFP were transformed into tobacco leaves, and chloroplast autofluorescence was used as a marker. After 2–4 d, leaf lower epidermal cells were observed with a confocal microscope (Nikon AX, Nikon, Shanghai, China) using stimulation at 488 nm for GFP and chloroplasts. Fluorescence collection wavelength is ~500–530 nm (GFP) and ~650–750 nm (chloroplast), respectively.

### 4.5. Plasmid Construction, Plant Transformation, and Screening

Soybean (Jilin 26) leaf cDNA was used as a template for PCR amplification with primers CGS2-101-F and CGS2-101-R (Appendix A). Two restriction enzyme sites, Sac I and Xba I, were introduced, respectively, into the primers. The amplified fragment was recombined into the binary vector pTF101.1 (AB303311.1) with the CaMV 35S promoter, between the Sac I and Xba I sites, resulting in a 35S: GmCGS2 construct: NOS construct. The construct was verified by DNA sequencing and then transformed into Agrobacterium strain EHA105 for soybean transformation. Transgenic soybeans were obtained using the Agrobacterium-mediated soybean cotyledon node transformation method as described before with modifications [67]. After sterilization with chlorine, soybean seeds were germinated for about 24 h on germination medium before the transformation, then the scratched cotyledons were immediately immersed in the Agrobacterium infection solution for about 2 h. After being placed on the cocultivation medium for about 4 days in a dark incubator, the explants were imbedded in shoot induction medium. Two weeks later, the explants were transferred to fresh shoot induction medium containing 8 mg·L^−1^ glufosinate-ammonium for about two weeks. Then, the explants were transferred to stemming medium containing 4 mg·L^−1^ glufosinate-ammonium. Next, the elongated shoots were transferred to rooting medium when they were about 3–4 cm tall. After root emergence, the T0 plants were transferred to flowerpots (vermiculite:soil, 1:1), and were retested using phosphinothricin acetyl transferase (PAT) test strips (Beijing Aochuang Jinbiao Biotechnology Corporation, Beijing, China) and PCR with genomic DNA in leaves as the template to detect the expression of bar and the target gene, respectively. To distinguish the expression of *GmCGS2* in the original genetic background, the upstream and downstream primers used to identify positive plants were designed in the vector sequence and the CDS region of *GmCGS2*, respectively. Three independent transgenic events, named OE3, OE4, and OE10, were produced (T1). cDNA from developing seeds resulting from the three transgenic events were used as templates to clarify the introduction efficiency of *GmCGS2* using the primer pairs listed in Appendix A. T3 transgenic soybeans were used for all experiments.

### 4.6. Quantifications of Amino Acid and Soluble Protein Contents

For free amino acid content determination, developing seeds at 15, 55, and 75 d (15D, 55D, and 75D) from the three transgenic lines and WT were harvested and ball-milled for 180 s to a fine powder. Free amino acids were extracted from the powdered samples (100 mg) by vortexing in the 4 mL of water. Aliquots (50 μL) were taken and homogenized with 200 μL of acetonitrile/methanol (1:1), which contained mixed amino acid internal standards. After centrifugation at 12,000 rpm for 10 min to remove solids, the supernatant was recovered, and 1 μL was analyzed using an ultra-high-performance liquid chromatography coupled to tandem mass spectrometry (UHPLC-MS/MS) system (ExionLC™ AD UHPLC-QTRAP^®^ 6500+, AB SCIEX Corp., Boston, MA, USA) equipped with an ACQUITY UPLC BEH Amide (2.1 × 100 mm^2^, 1.7 μm). The temperature of the column was maintained at 50 °C. The mobile phase, composed of solvent A (0.1% formic acid in 5 mM ammonium acetate) and solvent B (0.1% formic acid in acetonitrile), was used in gradient mode at a flow rate of 0.30 mL/min. The mass spectrometer was operated in positive multiple-reaction mode. The main parameters were as follows: ion spray voltage (5500 V), curtain gas (35 psi), and ion source temperature (550 °C). Amino acid standard curves were used for quantitation.

Dry seeds of three transgenic events and a control were used for hydrolytic amino acid extraction. Powdered samples were screened using a 60-mesh sieve, transferred to digestion tubes, and soluble protein was hydrolyzed using 10 mL of 6 M HCL and 100 μL of β-mercaptoethanol at 100 °C in a drying oven for 24 h. Hydrolyzed amino acids were subsequently quantified using an S-433D amino acid analyzer (Skyam, Eresing, Germany).

The same batches of dry seeds were also sampled for total soluble protein content determination. Powdered samples were screened using a 100-mesh sieve, and total protein was extracted using the tricarboxylic acid (TCA)/acetone method, as previously described [68]. Soluble protein was quantified using the Bradford method [69].

### 4.7. Library Preparation and RNA Sequencing

Transgenic line OE10 and WT developing seeds at 15D, 55D, and 75D were collected for RNA sequencing (RNA-seq) analysis. Library construction and quality inspection were carried out by Novogene Co., Ltd., (Beijing, China), and an Agilent 2100 Bio Analyzer system (Agilent Technologies, Santa Clara, CA, USA) was used to measure the insert size of the library. qRT-PCR was then used to quantify the effective concentration of the library accurately with primers listed in Appendix A. For Illumina sequencing, different libraries were pooled according to the requirements of effective concentration and target unload data amount. To ensure the quality and reliability of the data analysis, the original data were subjected to filtering, sequencing error rate check, and guanine–cytosine (GC) content distribution check, and the clean reads were used for subsequent analysis.

### 4.8. RNA-seq Data Analysis

For analysis of differential gene expression, DESeq2 software was used based on a negative binomial distribution model. The P-value was corrected using the Benjamini–Hochberg approach for controlling the proportion of false positives [70]. Genes analyzed by DESeq2 (*p* ≤ 0.05 and |log2(fold change)| ≥ 1) were identified as differentially expressed genes (DEGs).

Functional annotation and enrichment analysis were performed using the following databases: GO (Gene Ontology); KEGG (Kyoto Encyclopedia of Genes and Genomes); STRING (protein interaction database); and Plant TFDB 4.0 (plant transcription factor database). After taking the log2 (fpkm + 1) value and centralized correction, the expression levels of DEGs were clustered using the H-cluster method. Enrichment of the DEGs into metabolic pathways was determined using the KEGG database, and the metabolic pathways were considered statistically significant when *p* < 0.05. The binding sites of target gene promoter elements and differential transcription factors were predicted using PlantPAN 3.0 (http://plantpan.itps.ncku.edu.tw/, accessed on 15 May 2021).

### 4.9. Statistical Analysis

Principal component analysis (PCA) was performed using the gene expression level (fragments per kilobase of transcript per million mapped fragments, FPKM) of all samples to evaluate sample similarity and find correlating transcript changes, using metaX. [71]. Omic Studio tools (https://www.omicstudio.cn/tool, accessed on 10 May 2021) were used to calculate the Pearson’s correlations. All data obtained from this study were analyzed statistically with IBM SPSS 19 statistical software (SPSS Inc., Chicago, IL, USA) and Microsoft Excel 2007. Data were first tested for the heteroscedasticity of error variance and normality with Levene’s and Shapiro–Wilk tests. One-way ANOVA and Duncan’s Multiple Range Test were performed to assess differences in free and protein-bound amino acid content between the transgenic lines and the WT. Two-way ANOVA and Tukey’s test were carried out to assess the expression levels of genes, with time and genes as factors. Significance levels: ** *p* < 0.01, * *p* < 0.05.

## Figures and Tables

**Figure 1 ijms-24-14125-f001:**
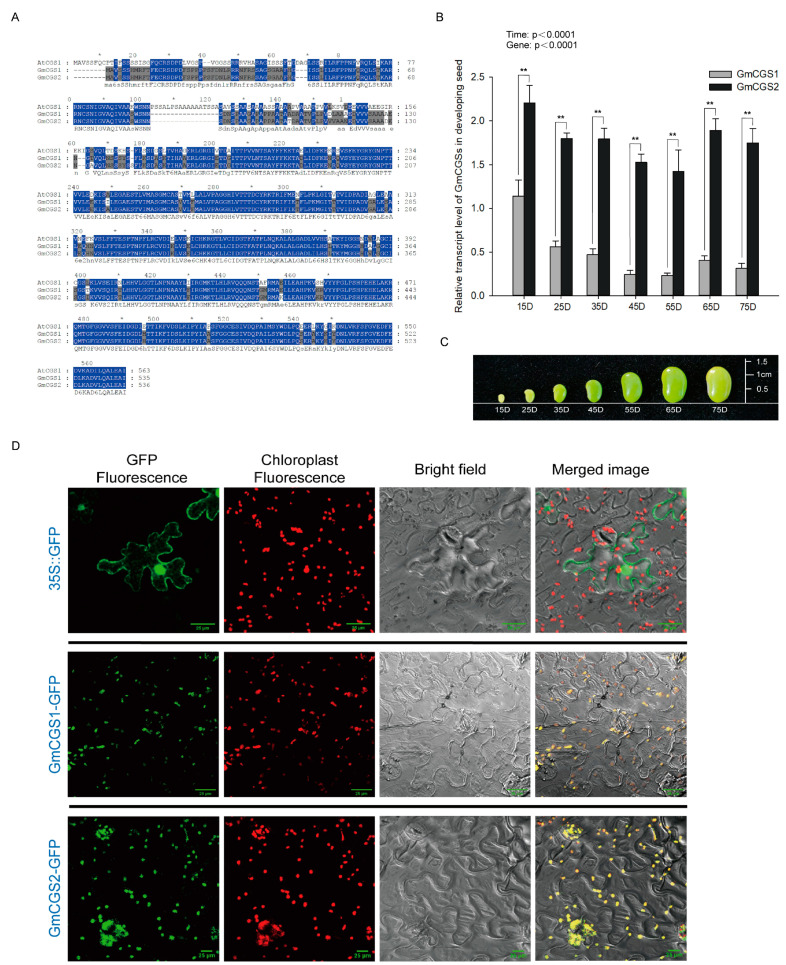
Features of the CGSs from soybean. (**A**) Alignment of amino acids of GmCGS1, GmCGS2, and AtCGS1, sequences were aligned using MEGA 6.0. (**B**) qRT-PCR analysis of *GmCGS1* and *GmCGS2* in developing seeds at 15 days, 25 days, 35 days, 45 days, 55 days, 65 days, and 75 days after flowering from soybean JACK. Values were normalized against the result for soybean Actin. The values shown represent the means ± SD (*n =* 6). The asterisks indicate the statistically significant changes from transgenic lines to the control (*, *p* < 0.05; **, *p* < 0.01; by two-way ANOVA). (**C**) Developing seeds at 15, 25, 35, 45, 55, 65, and 75 days after flowering. (**D**) Subcellular localization of GmCGS1 and GmCGS2 in tobacco leaves. Green represents the fusion protein fluorescence, and red represents chloroplast autofluorescence. Scale bar *=* 25 μm.

**Figure 2 ijms-24-14125-f002:**
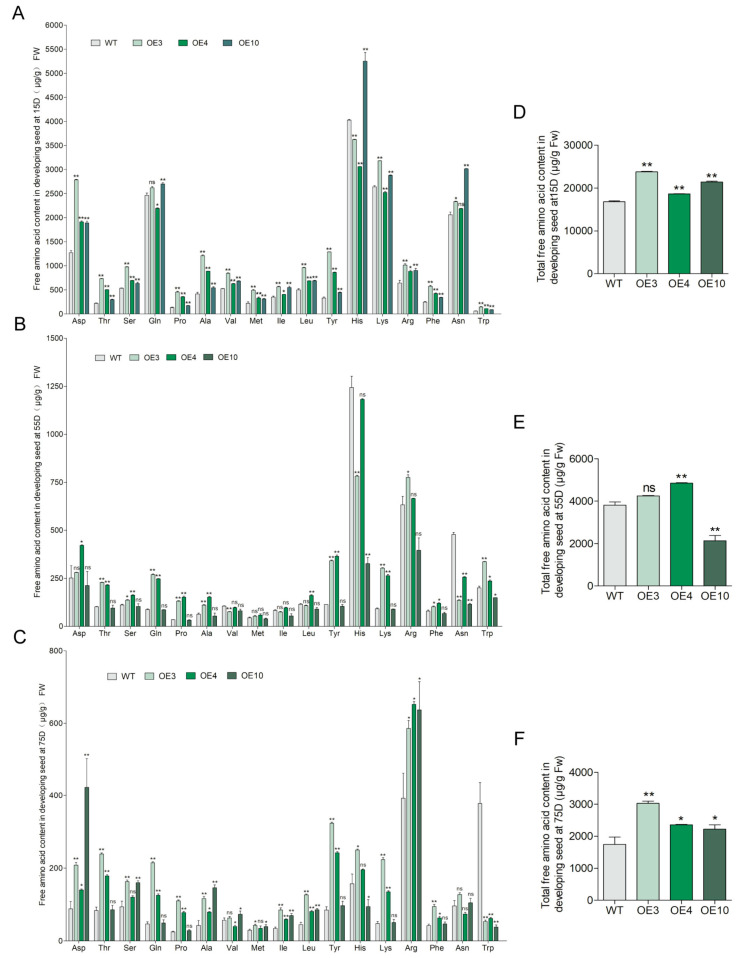
Free amino acids content in developing seeds at 15, 55, and 75D after flowering in three transgenic lines and WT. Amino acids were quantified with UHPLC-MS/MS, and their contents were calculated as μg per g fresh weight of developing seeds. Free and total amino acid content in transgenic lines and WT developing seeds at 15D (**A**,**D**), 55D (**B**,**E**), and 75D (**C**,**F**). The data shown represent the means ± SD (*n =* 6). The asterisks indicate the statistically significant changes from transgenic lines to the control (*, *p* < 0.05; **, *p* < 0.01, n.s.= not significant, one-way ANOVA followed by Duncan’s Multiple Range Test).

**Figure 3 ijms-24-14125-f003:**
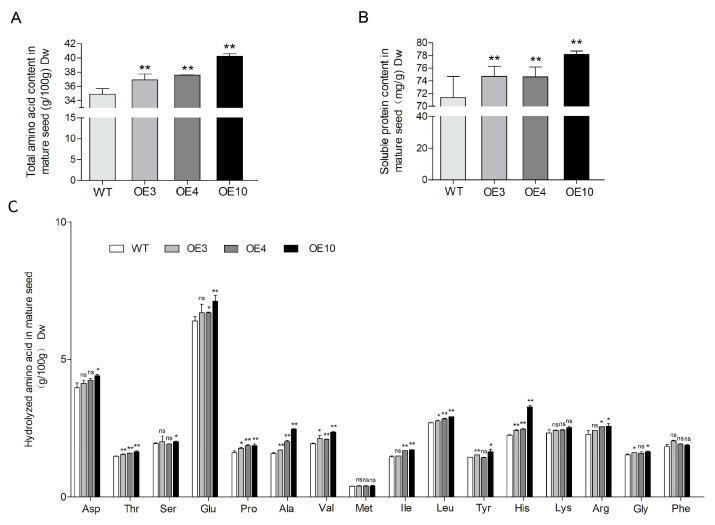
Overexpression of *GmCGS2* promotes the accumulation of total amino acids and soluble protein in soybean dry mature seeds of transgenic lines. (**A**) Total amino acid contents in seeds of transgenic lines and the control. (**B**) Soluble protein contents (extracted with TCA/acetone and measured by the Bradford method) in seeds of transgenic lines and the control. (**C**) Sixteen hydrolyzed amino acid contents in seeds of transgenic lines and the control. The levels of amino acids were detected by an amino acid analyzer after protein hydrolysis. Values are means ± SD (n *=* 6). The Asterisks indicate the statistically significant changes from transgenic lines to the control (*, *p* < 0.05; **, *p* < 0.01, n.s.= not significant, one-way ANOVA followed by Duncan’s Multiple Range Test).

**Figure 4 ijms-24-14125-f004:**
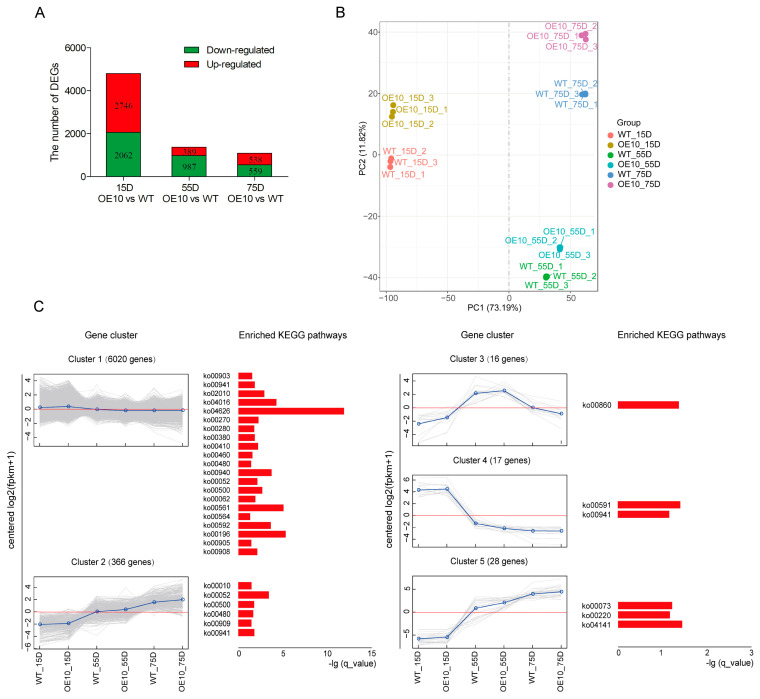
Overview of soybean transcriptome response to overexpress *GmCGS2* in developing seeds at 15D, 55D, and 75D. (**A**) Number of DEGs that were significantly up- or down-regulated at three developmental stages of developing seeds in OE10 compared to WT (*p* ≤ 0.05 and |log2(foldchange)| ≥ 1). (**B**) Principal component analysis plots based on the transcript levels identified by RNA-seq of WT and transgenic line OE10. (**C**) Cluster analysis of DEGs between WT and transgenic line OE10 at three time points based on the H-cluster method. The enriched KEGG pathways for each profile are listed on the right. Enrichment scores are shown as −lg (q_value).

**Figure 5 ijms-24-14125-f005:**
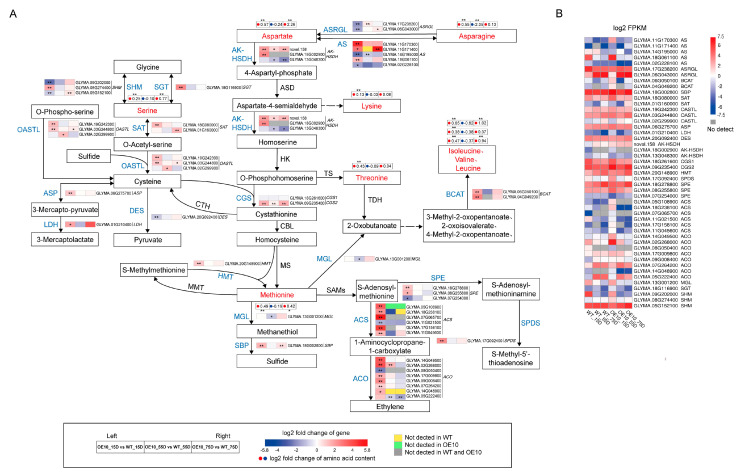
Expression patterns of DEGs and free amino acid contents related to methionine metabolism in developing seeds at 15D, 55D, and 75D of WT and OE10. (**A**) Pathway schematic. Log2 fold change of amino acid contents and DEGs in three comparison combinations are presented in associated pathways. Differentially expressed metabolites and DEGs are shown in red and blue text, respectively. Solid arrows represent biosynthesis steps established, while broken arrows indicate the involvement in a variety of enzymatic reactions. The yellow, green, and gray boxes indicate that the transcript is not detected in WT, OE10, and both of them in the corresponding comparison combination, respectively. The asterisks indicate the statistically significant changes from transgenic lines to the control (*, *p <* 0.05, **; *p <* 0.01). (**B**) Heat map of FPKM values of the DEGs involved in the schematic pathway. The gray boxes indicate that the transcript is not detected.

**Figure 6 ijms-24-14125-f006:**
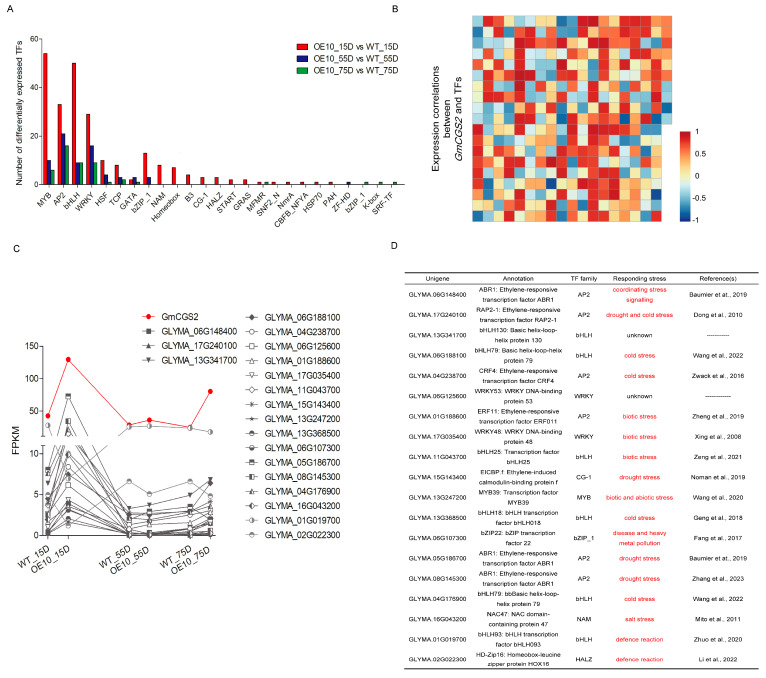
Screening of transcription factors coexpressed with *GmCGS2*. (**A**) Differently expressed transcription factors identified in three comparison combinations. All the transcription factors had been classified into various types, the number of transcription factors of which has been counted. (**B**) A Pearson’s correlation analysis between the expression levels of all differently expressed transcription factors and *GmCGS2* in all samples. (**C**) Transcription factors significantly correlated and possibly coexpressed with *GmCGS2* with a Pearson’s correlation coefficient ≥ 0.85 were screened. (**D**) Explanatory notes of transcription factors from (**C**). Baumler et at., 2019, [36]; Dong et al., 2010, [37]; Wang et al., 2022, [38]; Zwack et al., 2016, [39]; Zheng et al., 2019, [40]; Xing et al., 2008, [41]; Zeng et al., 2021, [42]; Noman et al., 2019, [43]; Wang et al., 2020, [44]; Geng et al., 2018, [45]; Fang et al., 2017, [46]; Zhang et al., 2023, [47]; Mito et al., 2011, [48]; Zhuo et al., 2020, [49]; Li et al., 2022, [50].

## Data Availability

Raw sequence data are available at the NCBI Sequence Read Archive under BioProject PRJNA924064.

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
