# Peer review of "Overexpressing *GmCGS2* Improves Total Amino Acid and Protein Content in Soybean Seed"

_ijms, 2023, doi:10.3390/ijms241814125_

Round 1
Reviewer 1 Report
The manuscript entitled “Overexpressing GmCGS2 improves total amino acids and protein content in soybean seed” by Zhang et al., reports overexpression of a GmCGS2 gene in soybean. The authors claim that the overexpression resulted in increased free Methionine as well other free AA and protein content in seeds.
Introduction
The intro first paragraph focused on role of Met in SAM. Its OK to add such detail but I think is less relevant to the objective of the MS. Major focus should be the Met content in seed and protein content in seed. Talk more specifically how Met content is related with soy protein, and especially which proteins’ content is associated with Met.
L57. Aspartic acid family pathway or “Aspartate-family pathway”? use the universally accepted terminology here.
The last paragraph of intro. Be consistent in italicizing the gene names. The same for the Results and discussion section.
The results contain several errors in presentation of scientific names of species. They aren’t italicized and if used second time shouldn’t be expressed as full genus name.
Though not a part of this manuscript, it would be more interesting to understand how the individual content of 11S and 7S proteins are affected in the transgenes since those are the major soy proteins.
The figures, especially those not involving graphs are too blur to give any detail. Authors should provide high resolution images.
In Figure 5. What does “not detected” mean? Is it not found in the WT using PCR or transcriptome seq or other at particular stage or not differentially expressed?
Discussion. On many instances, authors repeated their results+methods. This should be revised carefully. These are multiple so I cant refer to all of these one by one.
If free Met does not effect the content of major sulfur containing aa proteins proteins in seeds, then what is the benefit for overexpression? What nutritional benefits it can bring?
Discussion section should be shortened and avoid repetition and unnecessary debate. Major findings are the overexpression increases the free AA content and effects TPC. Then next is transcriptome related results. I suggest dividing the D section into two. First talk about what general changes did the overexpression resulted in. Then talk about how these changes were possibly governed.
English language is ok but authors must focus on italicizing sci names and gene abbreviations.
Author Response
Dear Editor
We appreciate the positive comments about the manuscript from you and reviewers.
We have checked the manuscript and revised it according to you and reviewer`s
suggestion. The marked versions of the revised manuscript were submitted. The line
numbers of the revisions are present in marked version. All responses to the concerns
indicated by the reviewers are detailed in the rebuttal letter. Please do not hesitate to
contact me if there are further issues to be clarified.
Sincerely,
Fan Yan,
Ying Wang

Reviewer 2 Report
Manuscript "Overexpressing GmCGS2 improves total amino acid and protein content in soybean seed" is very interesting.
General comments:
Authors clarified the mechanisms of regulation of soybean CGS on Met biosynthesis. Authors overexpressed soybean CGS 2 (GmCGS2) in the soybean variety “Jack” and carried out transcriptome sequencing analysis of seeds at different development stages. Authors examined the effect of gene manipulation on amino acid and protein content in developing and mature soybean seeds.
Detailed comments:
Introduction is very good.
Quality of Figure 1 is poor. Needs correction.
Figure 1: The authors caption Figure 1B as the results of a one-way ANOVA. The figure shows that the experiment was two-way ANOVA. Correction should be made. What about the interaction?
Figure 2: Similar to Figure 1: this is a two-factor experience. Correction should be made.
Figure 3C: Similar to Figure 1: this is a two-factor experience. Correction should be made.
My suggestion:
"log10": The nomenclature 'log10' does not require the notation '10'. It is a standard mathematical notation. It should be corrected.
The description of subsection "4.9 Statistical analysis" is very cursory. It should be supplemented with all the statistical methods used during the implementation of the assumptions. In addition, the experiment was assumed to be bivariate and should be analyzed as such. Statistical analyses based on the two-factor model should be repeated.
Paper needs major revision.
Author Response

(The authors gave the same response as above.)

Round 2
Reviewer 1 Report
Authors have addressed the comments and improved accordingly. no further comments
minor checks are required for final look.
Author Response
Dear editor
We appreciate the positive comments about the manuscript from you and reviewers. We have rechecked our manuscript and modified our grammar and description. Please do not hesitate to contact me if there are further issues to be clarified.
Sincerely,
Fan Yan,
Ying Wang
Reviewer 2 Report
The authors have not incorporated any(!) of the suggested amendments. Under these circumstances, the manuscript should be rejected.
Author Response
Dear Editor
We are very grateful for your valuable feedback. We have accepted your suggestions and made modifications, which may have caused misunderstandings due to our improper description. I'm very sorry, and the specific modifications are as follows. Please do not hesitate to contact me if there are further issues to be clarified.
Sincerely,
Fan Yan,
Ying Wang
Major comments:
The quality of figure 1 is still poor. It needs correction.
Response: Thanks for your suggestions. When we reupload our manuscript, we can only upload it in PDF format, and after converting the Word file to a PDF file, the resolution of the figures decreased. The quality of original figures is very high, and the uploaded compression package contains all the original figures. We also added the original image of Figure 1 in the attachment uploaded below the reply dialog box.
Figure 1: The authors replaced one word in the caption of Figure 1B, while nowhere in the manuscript are the results of the two-way analysis of variance described. The manuscript should be supplemented with a description of these results and then discussed.
Response: We are very grateful to your suggestions on our manuscript, and “We found that GmCGS1 and GmCGS2 exhibited similar trends in their expression patterns, first decreasing by 79.6 % and 35.5 %, respectively, from 15D to 55D, then in-creasing by 74.6 % and 33.1 %, respectively, at 65D, and finally decreasing again by 22 % and 7.4 %, respectively, at 75D (Figure 1B). Despite the highly similar subcellular local-ization, amino acid sequences, and expression patterns shared by GmCGS1 and GmCGS2, we noticed that the level of expression of GmCGS2 was higher than that of GmCGS1 at all developmental stages.” was modified to “We found that the content of the expression of GmCGS1 and GmCGS2 exhibited similar change trends, and were influenced significantly by the developmental stages, that were first decreasing by 79.6 % and 35.5 %, respectively, from 15D to 55D, then in-creasing by 74.6 % and 33.1 %, respectively, at 65D, and finally decreasing again by 22 % and 7.4 %, respectively, at 75D (F (6, 112) =5.729, p< 0.0001) (Figure 1B). This result suggested that despite the highly similar subcellular localization, amino acid sequences, and expression patterns shared by GmCGS1 and GmCGS2, we noticed that the level of expression of GmCGS2 was significantly higher than that of GmCGS1 at all develop-mental stages (F (6, 112) =84.57, p< 0.0001).” in line 115-123.
Figure 2: Similar to Figure 1: this is a two-factor experiment. A correction should be made.
Figure 3C: Similar to Figure 1: this is a two-factor experiment. A correction should be made.
Response: We are very appreciating the reviewer’s suggestions. In Figure 2, we aim to compare the transgenic lines with the control independently. Similar to Figure 3C, we want to compare the differences between the transgenic lines and the control in mature seeds. After discussing with other authors, we think that Student’s t test is more suitable for analyzing the results described in Figure 2 and Figure 3C. Besides, in order to make our statement more accurate, Figure 2 has been separated into three small graphs respectively in line 169.
The abscissa in Figure 2 and Figure 3C represents the division of different kind of amino acids, and is not a variable in the experimental process. We have reanalyzed the results of Figure 2 and Figure 3C, and modified the relevant descriptions in our manuscript. Similar statistical analysis was also presented in previous studies, such as the research by Wang et al., and Lu et al., [1-2].
[1] Wang, J., Wu, B., Lu, K., Wei, Q., Qian, J., Chen, Y., Fang, Z. The Amino Acid Permease 5 (OsAAP5) Regulates Tiller Number and Grain Yield in Rice. Plant Phy. 2019, 180, 1031-1045. doi: 10.1104/pp.19.00034.
[2] Lu, K., Wu, B., Wang, J., Zhu, W., Nie, H., Qian, J., Huang, W., Fang, Z. Blocking amino acid transporter OsAAP3 improves grain yield by promoting outgrowth buds and increasing tiller number in rice. Plant Biotechnol J. 2018, 10, 1710-1722. doi: 10.1111/pbi.12907.
The description of subsection "4.9 Statistical analysis" is still very sketchy. It should be supplemented with all the statistical methods used during the assumptions. In addition, it was assumed that the experiment is a two-factor experiment and should be analyzed as such. There is a lack of information about checking the assumptions about the feasibility of ANOVA.
Response:
We are very grateful to your comments on our manuscript, and we are very sorry for the lack of information about checking the assumptions about the feasibility of ANOVA. We have supplemented the description of subsection "4.9 Statistical analysis" in line 559-565 as “All data obtained from this study were analyzed statistically with IBM SPSS 19 statis-tical software (SPSS. Inc., Chicago, IL, USA) and Microsoft Excel 2007. Data were first tested for the heteroscedasticity of error variance and normality with Levene’s and Shapiro-Wilk tests. Student t-tests were performed to assess differences between the transgenic lines and the WT. Two-way ANOVA and Tukey's test were carried out to assess the expression levels of genes, with time and genes as factors. Significance levels: **p < 0.01, *p < 0.05.”
A description of some statistical methods can be found in other sections of the manuscript. This should definitely be corrected.
Response: Thank you very much for bringing up this issue to our attention. We have carefully rechecked our manuscript, and the description of some statistical methods (PCA analysis and Pearson’s correlations) has been moved to the ‘4.9. Statistical analysis’ section in the revised manuscript.

Round 3
Reviewer 2 Report
The quality of figure 1 is still poor. It needs correction.
Figure 1: The authors replaced one word in the caption of Figure 1B, while nowhere in the manuscript are the results of the two-way analysis of variance described. The manuscript should be supplemented with a description of these results and then discussed.
Figure 2: Similar to Figure 1: this is a two-factor experiment. A correction should be made.
Figure 3C: Similar to Figure 1: this is a two-factor experiment. A correction should be made.
The description of subsection "4.9 Statistical analysis" is still very sketchy. It should be supplemented with all the statistical methods used during the assumptions. In addition, it was assumed that the experiment is a two-factor experiment and should be analyzed as such. There is a lack of information about checking the assumptions about the feasibility of ANOVA.
A description of some statistical methods can be found in other sections of the manuscript. This should definitely be corrected.
Paper needs major revision.
Author Response

(The authors gave the same response as above.)

Round 4
Reviewer 2 Report
The low quality of Figure 1. The explanation given by the authors that the quality of the graph is due to the conversion of the file is very strange. Figures can be converted as, for example, .eps and then they do not lose quality.
Figure 1: No description of the results of the two-way analysis of variance. The answer that "It was found that GmCGS1 and GmCGS2 showed similar trends in their expression patterns" should be confirmed with appropriate analyses. This is what I have been reporting all along in the previous two rounds of reviews.
Figure 2: similar to Figure 1.
Figure 3: similar to Figure 1.
Author Response
Dear Editor
We have checked the manuscript and revised it according to you and reviewer`s suggestion. The line numbers of the revisions are present in marked version. All responses to the concerns indicated by the reviewers are detailed in the rebuttal letter. Please do not hesitate to contact me if there are further issues to be clarified.
Sincerely,
Fan Yan,
Ying Wang
Major comments:
The low quality of Figure 1. The explanation given by the authors that the quality of the graph is due to the conversion of the file is very strange. Figures can be converted as, for example, .eps and then they do not lose quality.
Response: Thanks for your comments, and we have modified the format of Figure 1 to maintain its good quality in PDF format file.
Figure 1: No description of the results of the two-way analysis of variance. The answer that "It was found that GmCGS1 and GmCGS2 showed similar trends in their expression patterns" should be confirmed with appropriate analyses. This is what I have been reporting all along in the previous two rounds of reviews.
Response: We are very grateful to your suggestions on our manuscript, and the description of the results of the two-way analysis, “The expression levels of these two genes were influenced significantly by the developmental stages (F (6, 72) =22.83, p< 0.0001), gene types (F (1, 72) =775.1, p< 0.0001), and their interactive effect (F (6, 72) =6.805, p< 0.0001).” and “The expression levels of GmCGS1 and GmCGS2 were first decreased by 79.6 % (p< 0.0001) and 35.5 % (p< 0.0001), respectively, from 15D to 55D, then increased by 74.6 % (p=0.9020) and 33.1 % (p< 0.0001), respectively, at 65D, and finally decreased again by 22 % (p=0.9972) and 7.4 % (p< 0.0001), respectively, at 75D” was added in line 116-118, 123-127. The expression levels of GmCGS1 and GmCGS2 in developing seed at 15-75D were analyzed with the Pearson’s correlation, and the relevant description was added in line 118-123.
Figure 2: similar to Figure 1.
Figure 3: similar to Figure 1.
Response: We are very appreciating the reviewer’s suggestions. In Figure 2A, 2B, 2C and Figure 3C, we focused on the differences before and after transgenic transformation including various free amino acid contents in immature seeds and various bound amino acid contents in mature seeds, and there is only one factor in each figure. The data of different amino acids such as aspartate, methionine, glutamate, et al. was measured independently, and we combined each single amino acid into one figure (2A, 2B,2C and 3C) to present better visual effects. Besides, whether genetically modified or not, there are differences in the content of different types of amino acids inherent, which is not the focus of our experiment. Therefore, Figure 2A, 2B, 2C and Figure 3C have only one purpose, that is to demonstrate whether there are differences in the target traits between the transgenic lines and the control, for further revealing the function of the gene. This is a single factor comparison experiment, and that Student’s t test is more suitable to analyze the results described in Figure 2A, 2B, 2C and Figure 3C. Statistical analysis similar to Figures 2A, 2B,2C and Figure 3C was also presented in previous studies [1-2]. Besides, the descriptions of the results of the student’s t test were added in line 162-163, 164, 171-172, 174-175, 178-179,196-197, 202-204.
[1] Liu, S., Wang, D., Mei, Y., Xia, T., Xu, W., Zhang, Y., You, X., Zhang, X., Li, L., Wang, N. Overexpression of GmAAP6a enhances tolerance to low nitrogen and improves seed nitrogen status by optimizing amino acid partitioning in soybean. Plant Biotechnol J. 2020, 8, 1749-1762. doi: 10.1111/pbi.13338..
[2] Xiang, X., Wu, Y., Planta, J., Messing, J., Leustek, T. Overexpression of serine acetyltransferase in maize leaves increases seed-specific methionine-rich zeins. Plant Biotechnol J. 2018, 5, 1057-1067. doi: 10.1111/pbi.12851.